# Severe Acute Ischemia of Glans Penis after Achieving Treatment with Only Hyperbaric Oxygen Therapy: A Rare Case Report and Systematic Literature Review

**DOI:** 10.3390/jpm13091370

**Published:** 2023-09-12

**Authors:** Adrian Hașegan, Ionela Mihai, Dan Bratu, Ciprian Bacilă, Mihai Dan Roman, Cosmin Ioan Mohor, Adrian Teodoru, Sabrina Birsan, Cosmin Mutu, Călin Chibelean, Maria Totan, Nicolae Grigore, Gabriela Iancu, Horatiu Dura, Adrian Boicean

**Affiliations:** 1Faculty of Medicine, Lucian Blaga University of Sibiu, 550169 Sibiu, Romania; adrian.hasegan@ulbsibiu.ro (A.H.); dan.bratu@ulbsibiu.ro (D.B.); ciprian.bacila@ulbsibiu.ro (C.B.); mihai.roman@ulbsibiu.ro (M.D.R.); cosmin.mohor@ulbsibiu.ro (C.I.M.); adrian.teodoru@ulbsibiu.ro (A.T.); sabrinaandreea.marinca@ulbsibiu.ro (S.B.); cosmin.mutu@ulbsibiu.ro (C.M.); maria.totan@ulbsibiu.ro (M.T.); nicolae.grigore@ulbsibiu.ro (N.G.); gabriela.iancu@ulbsibiu.ro (G.I.); horatiu.dura@ulbsibiu.ro (H.D.); adrian.boicean@ulbsibiu.ro (A.B.); 2Faculty of Medicine, University of Medicine and Farmacy, 540139 Targu Mures, Romania; calin.chibelean@umfst.ro

**Keywords:** circumcision, hyperbaric oxygen therapy, penile glans, ischemia, urology, complications, case report

## Abstract

Acute ischemia of the glands is a severe complication after circumcision. We outline the challenging case of a seventeen-year-old boy with glandular ischemia (GI) that appeared shortly after circumcision. Methods: We present a case report and literature review related to glans ischemia, and the complications of circumcision are reviewed. We note that there are very few cases described in the literature. Our patient was successfully treated with hyperbaric oxygen therapy (HBOT) after four days of no positive effect after all medical and surgical treatments written in the literature: Subcutaneous enoxaparin, local application of a glyceryl trinitrate, continuous epidural perfusion, intravenous pentoxifylline, alprostadil, intraoperative drainage, and aspiration with saline solution and epinephrine. Clinical improvement was noted at the first session of HBOT. A number of days after the operation, the penis looked normal and was healing. Complete healing of the penile glans was successfully realized one month after surgery. Conclusion: Based on the review and the case presented, we conclude that HBOT is the treatment of choice for acute ischemia of the penile glans, especially when other treatments do not work.

## 1. Introduction

Circumcision represents a surgical procedure involving taking away the foreskin wrapping the glans, and it represents one of the most common surgical procedures performed on pediatric patients [1,2]. The procedure is statistically associated with about a 1% rate of possible complications. The most common complications cited in the literature are bleeding, swelling, cosmetic deformities, and wound infection [3]. Special cases of complications in hemophiliacs deserve special mention, as circumcision in these cases may increase the potential for complications such as severe bleeding resulting in hematomas that can lead to acute ischemia [1,2,3]. We note that in practice, the most common type of disease encountered is hemophilia type A. Hemophiliac patients may present uncontrolled bleeding during the surgery, which demands special therapeutic management. Studies show that although factor VIII concentrate is administered before surgery, voluminous hematoma is difficult to deal with and can lead to ischemia as a result of compression. For hemophiliac patients, administration of factor VIII concentrate should be performed within eight or twelve hours to limit complications [1,2,3].

Even though circumcision is a well-standardized type of surgery with a well-established operating protocol, in some cases, major complications may occur during the procedure. Some of the possible complications noted in the guidelines are glans or penile amputation due to necrosis, infections, and even septicemia, and rare cases of urethrocutaneous fistulas are noted in studies [4]. Glans ischemia (GI) occurring after circumcision is a very severe complication, even if it presents a low incidence; it may even be life threatening, especially for pediatric patients [5]. Furthermore, the precise etiology that leads to ischemia is still uncertain, and there are no specialized guidelines for treatment due to the limited number of cases in the literature. Further studies of the etiology should be conducted to establish the risk factors implicated in acute ischemia after circumcision. Studies outline different ischemia causes, such as vasospasm of the glans after surgery, or another cause could be dorsal penile nerve block anesthesia due to vascular obstruction and edema after injecting the anesthetic solution during the surgery for circumcision. Another explanation of acute GI considered after surgery is hematoma and edema at the injection site that may lead to ischemic endothelial injuries and necrosis of the glans. Moreover, we outline that studies show that, in the case of a lack or delay of treatment, complications can arise, such as irreversible necrosis with major issues, such as irreversible loss of the physiological functions of the penis [5].

Many studies have reported glans ischemia (GI) in pediatric patients as a major complication after circumcision. We note that this pathology may be life threatening in children, and pediatric patients may suffer severe and irreversible complications, even penis amputation. Although ischemic complications of the glans penis occurring after circumcision represent a rare complication, they can lead to potentially irreversible necrosis, and studies also report severe long-term consequences for the patient, representing a life-threatening major complication, especially in surgeries realized in neonates [4,5,6,7,8,9].

Studies advise personalized, accurate treatment and that the first aim of proper management should be to provide a proper blood supply and also improve the oxygen distribution to the ischemic organ. Guidelines present different therapeutic alternatives, which include using hyperbaric therapy (HBOT) in order to provide sufficient oxygen to the ischemic organ, vasodilators such as pentoxifylline (PTX), and anticoagulants such as enoxaparin; other therapeutic alternatives are iloprost, corticosteroids, and peridural anesthesia [6,7,8,9,10,11,12,13,14]. Other studies report glans ischemia as a result of circumstantially constricting sutures and compressive surgical bandages; some studies describe the use of monopolar diathermy as an etiological factor.

Although the literature is limited regarding cases of glans ischemia and proper case management, studies have reported that surgeons should avoid exerting dorsal penile nerve block (DPNB) or administering vasoconstrictor drugs for local anesthesia [7]. However, even if these aspects are noted as a frequent cause of acute ischemia of the penile gland, a case report of glans ischemia in a surgery that did not use local anesthesia has also been reported [7].

Therapeutic methods that lead to perfusion of the ischemic organ are described in a few case reports and include the administration of vasodilation drugs, such as topical nitroglycerin or hormonal therapy with testosterone. There are also some case reports that note therapeutic success after using an epidural infusion of bupivacaine. The explanation for the caudal infusion of bupivacaine is that it decreases sympathetic innervation, and the infusion of bupivacaine increases the arterial supply, resulting in a better perfusion with an increased oxygen supply. It also improves venous drainage of the penile gland, and guidelines suggest it as a reliable therapeutic alternative that is also used in monotherapy [10] but with delayed reperfusion, as compared to other therapeutic methods that involve an association with intracavernous injection of glycerol trinitrate, which improves oxygen perfusion by achieving post-arterial smooth muscle relaxation [7,8,11].

We note that other reviews of case reports describe the use of corticosteroids in association with hyperbaric oxygen therapy as a successful way to improve oxygen delivery to the ischemic glans and save the organ [7,8].

Efe et al. outlined, in their study, a high D-dimer level in cases of vascular penile thrombosis, diagnosed clinically and via ultrasonography, with normal values of D-dimer after five days of anticoagulation with enoxaparin treatment, and after treatment, power Doppler showed normal blood flow of the penis [9]. According to this study and other research, we note that anticoagulant therapy using enoxaparin is a good medical alternative in the case of GI after circumcision [5].

In cases of GI to prevent septic complications, taking into consideration that studies note the susceptibility of urinary infection (Gram-negative bacteria) and urological wound infection (Gram-positive/negative bacteria), these pathogens are well covered with the combination of amoxicillin/clavulanic acid, showing a low resistance to E. coli and Staphylococcus aureus haemolyticus [15,16,17,18,19,20,21,22,23,24]. We note that the overuse of antibiotics may result in microbiota alterations and even complications in cases of Clostridioides difficile colitis [12,15,16,17].

Our primary outcome is to highlight the therapeutic effects of hyperbaric therapy (HBOT) in treating acute glans penis ischemia after circumcision in a young teenager. We note that, in this case, this therapy improved the patient’s symptoms and was lifesaving. Our secondary outcome is to describe other rare cases presented in the literature and emphasize the importance of an accurate and early diagnosis in the case of acute glans penis ischemia after circumcision, as well as prompt and personalized case management.

## 2. Materials and Methods

We present the case of a teenage boy who developed acute glans penis ischemia after circumcision, performed under general anesthesia with dorsal nerve penile block and successfully treated with hyperbaric oxygen therapy (HBOT).

We also performed a systematic search on Google Scholar, PubMed, and the articles from the Web of Science databases. The date of the search was for one year, starting June 2022 to June 2023. Due to the limited studies on this pathology, we included studies and case reports published until June 2023. introducing the following terms: “glans ischemia”, “glans ischemia after circumcision”, “circumcision”, “glans ischemia ethology”, “complications of glans ischemia”, “hyperbaric oxygen therapy”, “penile glans ischemia”, “penile artery”, “penile artery vascular flux”, “penile necrosis”, and “complications after circumcision” In order to diminish the potential bias, two independent authors manually screened the studies and case reports; the agreement ratio was over 97% for the studies included and over 95% for the excluded studies. For the final decision, the authors discussed and consented to their decision. We considered a review based on PRISMA (the Preferred Reporting Items for Systematic Reviews and Meta-Analysis) guidelines, according to the following inclusion and exclusion criteria.

Inclusion criteria were peer-reviewed articles that described acute ischemia of the penis glans. We focused on articles describing ischemia after circumcision, treated in monotherapy with hyperbaric oxygen therapy; we also included articles that used HBOT in association with other drugs such as pentoxifylline and other drugs leading to vasodilation. We highlight the fact that there are very few case reports described in the literature and that hyperbaric therapy (HBOT) presents clinical benefits that can improve patients’ general condition and save the penis glans with a very rapid effect. Exclusion criteria were represented by non-peer-reviewed articles, duplicates, or other issues that led to ischemia; we also excluded patients with congenital vascular abnormalities. 

## 3. Results and Discussion

In our systematic review of the literature, the database search identified 493 records, including 185 duplicates. A median of 212 articles were selected for screenings; a total of 90 of them were excluded, and another 122 were considered for retrieval. A total of 55 articles were considered eligible, and 27 articles were analyzed in the review; 22 were original case reports and 5 were studies (Figure 1).

We note that, although the literature is very scarce concerning accurate treatment in cases of penile ischemia, in the studies, hyperbaric therapy was the best option, presenting many clinical benefits for the patients and encountering success as monotherapy or in association with other drugs. Due to the fact that hypoxia may interfere with and aggravate many urological diseases, even leading to acute ischemia refractory to other treatments, we outline the clinical benefits of hyperbaric therapy in preventing hypoxia. Furthermore, hyperbaric therapy has also been applied to other urological diseases, such as radiation-induced hemorrhaging cystitis, in which case reduced hematuria, Fournier gangrene, acute kidney injury, and, in this case, hyperoxia resulting from hyperbaric therapy lead to improved organ oxygen supplies and wound healing through its anti-inflammatory therapeutic effects. However, further research is needed to establish protocols for hyperbaric therapy. In some extreme cases, it may be lifesaving for patients and should be considered an option for practitioners. The etiology of glans ischemia is not fully understood, and at the moment the principal factors considered are gland edema, vasoconstriction due to anesthesia, and hematoma. Another cause could be dorsal penile nerve block anesthesia due to vascular obstruction, but the rarity of this complication raises other important theories such as genetics or individual factors. We highlight that at the moment, hyperbaric oxygen therapy is considered an adjuvant therapy; this review and case presentation outline its importance as a first line of therapy. We note that through HBOT, many cases report rapid and successful treatment in cases of acute ischemia, which represents a real-time race for practitioners. Polak et al. described, in their study, two neonate patients successfully treated with HBOT after circumcision. Male circumcision is an old and standard surgical procedure performed for many different reasons, from religious reasons to medical conditions. Various complications resulting after circumcision surgeries are highlighted in the literature, such as bleeding and infection, although glans ischemia after circumcision is reported with more frequency in the pediatric literature, with a very limited number of cases for adult patients; moreover, we note that the etiopathogenesis of glans ischemia is not elucidated, and it seems to be plurifactorial.

Migliorini F et al. noted the effective clinical treatment associated with HBOT and PTX in a male patient who was 24 years old. Pentoxifylline is a hemorheological agent that improves erythrocyte flexibility and has been considered important in reducing blood viscosity, which leads to improved tissue perfusion and microcirculation in the ischemic organ. In the case of this patient, this hypoxia provided by hyperbaric therapy in association with improved tissue perfusion by PTX represented the best option to prevent penis glans amputation [6]. Other studies also reported a successful outcome by administering pentoxifylline (PTX) as monotherapy or in association with multiple therapeutic methods [7]. Pepe P. et al., in their study, used HBOT with corticosteroids, and they recorded a fast recovery for their patient with the remission of acute symptoms and improved vascular flux and tissue perfusion. We note that HBOT is the common medical therapy for improving the vascular flux and preventing complications in the case of acute glans ischemia [17]. Pepe P et al. described a case of acute penis ischemia five days after undergoing circumcision. Upon his arrival at the hospital, the glans presented edema and a necrotic appearance. They decided to immediately treat him with antibiotics, antiplatelets, corticosteroids, and HBOT, with a total recovery of the patient and without any other complications [19].

Hyperbaric oxygen therapy (HBOT) has been applied to urological tissue healing, taking into account that it decreases inflammation, improves tissue regeneration through endothelial proliferation, stimulates tissue oxygen perfusion, and exerts bactericidal effects [13]. Hyperbaric oxygen therapy (HBOT) has been used with success in patients undergoing urological reconstructive surgery due to the fact that it improves graft integration. HBOT presents clinical benefits by accelerating tissue regeneration by increasing oxygen levels and improving angiogenesis and tissue reperfusion [14,15,16,17,18,19,20,21,22,23,24,25,26,27]. Pablo Garrido-Abad et al. described acute ischemia of the penis in a 3-year-old boy after circumcision. They noted different ischemia causes, such as vasospasm of the glans after surgery; another cause could be the dorsal penile nerve block due to vascular obstruction and edema after injecting the anesthetic solution during the surgery for circumcision. Another explanation that they suggest for GI after surgery is hematoma and edema at the injection site that may lead to ischemic endothelial injuries and necrosis of the glans. The patient presented a resolution of acute symptoms after hyperbaric oxygen therapy after the first session of therapy, followed by permanent improvement with each administration. Moreover, at the follow-up visit, the vascular flux on color Doppler ultrasonography was normal. Furthermore, the patient presented physiological functions of the glans and did not encounter difficulty in urination or urinary inconsistency [24]. In all cases, rapid and aggressive treatment was needed in order to preserve the normal functions of the organ, which represented an important challenge for practitioners. Studies recorded the most rapid resolution using hyperbaric oxygen therapy in monotherapy as well as in association with other drugs [6,17] (Table 1).

On the other hand, in studies that used other drugs before initiating hyperbaric oxygen therapy, remission of acute symptoms required more time, which endangered the restoration of normal organ function. One of the main concerns of practitioners in acute glans ischemia is restoring the normal function of the organ and preventing further complications. Guidelines outline a faster resolution of acute ischemia that diminishes the potential for organ dysfunction and decreases the number of complications, including infections and potential sepsis. 

We outline that, although limited in the literature we researched, of all the articles found, only six authors included HBOT in the treatment of penile glans ischemia as the only treatment (Polak et al. [17]) or in combination with (Pentoxifylline) PTX (in the studies realized by Migliorini F et al. [6], Elemen L et al. [25], Tzeng Y. et al. [18]). All the presented studies recorded rapid resolution of necrosis and ischemia after hyperbaric oxygen therapy. They also noted that, besides the apparition of necrosis, intensive treatment resulted in arterial flux restoration with multiple clinical benefits for patients. They also recorded the normal function of the organ after ischemic resolution and at follow-up visits. We highlight that the presented literature played a crucial role in the clinical management of the 17-year-old patient who presented with glans ischemia following circumcision for congenital phimosis and was treated in our clinical service [17,18,19,20,21,22,23,24,25,26,27,28,29].

## 4. Case Presentation

We present the case of a 17-year-old boy who underwent circumcision at our urological department for congenital phimosis. We note that the patient presented no other medical history other than this congenital phimosis that led to recurrent infections and difficulty retracting the foreskin. The surgical procedure was performed under general anesthesia with a dorsal nerve penile block of 10 mL of 0.5% bupivacaine but without using epinephrine (we note that we avoided using epinephrine especially to prevent possible vasospasm and potential ischemia).

The surgery was uneventful, with good homeostasis using bipolar diathermy. The skin was closed with interrupted 4-0 absorbable stitches. A circular dressing of the glans combined with a greasy ointment was applied (we outline that we considered that the circular dressing could have been causing potential pressure aggravating the edema and leading to ischemia of the penile artery), but during the surgery, there were no signs of ischemia and we did not have any intra-surgery issues of concern. At the end of the procedure, the glans appeared normally perused. We could not identify hematoma or edema at the injection site; the glans appeared well irrigated after the surgery, and the normal appearance of the mucosa was noted. However, five hours after surgery, an ischemic appearance of the glans was noticed without pain or difficulty urinating; symptoms aggravated during the first day (Figure 2A,B).

Blood analyses (coagulation test, antithrombin III level, protein C and S) were checked and found to be normal. A color Doppler echography (CDE) showed normal flow in the dorsal penile artery and no obstruction in the proximal vessel. However, the patient presented with acute ischemia of the glans and soon developed signs of necrosis. Aggressive treatment was started immediately. The treatment included first anticoagulant therapy, which was administered with enoxaparin 4000 UI daily during the hospitalization, and a topical patch of 25 mg of glyceryl trinitrate was applied to the glans in order to achieve vasodilatation, continuing with regular topical nitroglycerin 2% ointment twice daily; meanwhile, monitoring blood pressure and glans vascularity every two hours was performed. The constant monitoring showed a worsening of the constants, with the patient developing leukocytosis and increased PCR. Urinary summary was analyzed; however, we started the wound and urinary infection prophylaxis using amoxicillin 30 mg/kg for five days, associated with probiotics and prebiotics, in order to prevent microbiota dysbiosis or infection with Clostridioides difficile. Despite the intensive treatment according to guidelines, the patient did not present any improvement in the symptoms, and the necrosis persisted. The following day, a continuous epidural infusion of 0.2% ropivacaine and intravenous pentoxifylline (10 mg/kg, 100 mg) three times a day was started. A urinary catheter was inserted after the epidural in order to prevent failure to void after anesthesia. Unfortunately, in the case of our patient, no clinical benefit was observed after 48 h of intensive treatment (Figure 3). On the morning of the second postoperative day, intraoperative drainage and aspiration of the penile glans with saline solution and epinephrine (1/10.000 dilution) were performed in order to improve perfusion of the penile gland. Despite the intensive treatment and motorization, no significant improvement in the penile glans color or blood flow was noticed in the following days.

Prophylaxis of a wound and urinary infection was performed using amoxicillin/clavulanic acid (amoxicillin 30 mg/kg) for five days. Prostanoids (iloprost and prostaglandin (PGE1)) are in a class of medications called vasodilators. Alprostadil is a synthetic analog of prostaglandin E1. It works by relaxing the muscles and blood vessels, causing vasodilation (increasing peripheral blood flow, helping with erectile dysfunction), bronchodilation, and inhibiting platelet aggregation [6]. We used alprostadil 20 mg twice daily, hoping for a vasodilation response; unfortunately, the ischemia was refractory to the drug lines presented in the guidelines. Taking into account that, after five days, the usual therapy showed no therapeutic benefits for our patient, we referred the patient to an out-ambulatory department to start treatment with HBOT. We note that the signs of good perfusion of the penile glad were noticed after the first session of HBOT (90 min duration, oxygen pressure until 240 Kpa, at 100% O2) (Figure 2). The rapid resolution of acute ischemia after hyperbaric therapy strengthens the important clinical benefits of this treatment in cases of acute ischemia, and its role in urological diseases should be clearly defined by further studies (Figure 3).

We also outline that the psychological state of the patient improved significantly after treatment and following the remission of acute symptoms. Our case emphasizes the importance of adapting the therapeutic management of a rare complication following circumcision, leading to acute ischemia in the case of a young man.

Regarding our patient, the decision for treatment with HBOT was taken as the last option after four days of ineffective treatment. HBOT was very efficient from the first session, significantly improving the glans color and the patient’s mental state. We emphasize that hyperbaric therapy presents multiple therapeutic effects in the case of acute ischemia, leading to symptom resolution, such as inflammation reduction, endothelial proliferation that leads to angiogenesis, improved oxygen delivery, bactericide effects, preventing wound infection, and restoring normal vascular flux in cases of ischemia.

Our presented case showed normal penile and glandular flow at CDE (color Doppler echography) and a normal level of D-dimer. A total of five daily sessions of HBOT were performed. The patient had to continue only with anticoagulant therapy for a total of 5 days; meanwhile, all the other treatments were stopped, and the urinary catheter was removed. The patient was discharged on day seven postoperatively. At the one-month follow-up, the penis and glans were found to be in normal status, and we also noted that the patient presented normal function of the penis (Figure 4).

In light of the review presented and the case report we described, we highlight the importance of personalized case management and early diagnosis of potential complications in order to prevent irreversible glans, penile amputation, or septicemia.

In our patient, the anesthesia was achieved by using general anesthesia combined with a dorsal penile block used for perioperative analgesia; during the surgery, we used bipolar electrocautery. When the procedure was completed, we used a combination of antibiotic and corticosteroid ointment on the coronal suture. The penis was gently covered with gauze but without any tight circumferential bandages. We note that, as previously stated, the cause of developing acute ischemia might have been the association of the dorsal penile block used for analgesia, which could have led to edema and vasospasm. We emphasize that, in accordance with other case reports that encountered the same acute pathology, hyperbaric oxygen therapy represents a low-cost therapy that has very few adverse effects. Its clinical benefits, as well as its innovative and fast therapeutic effects, make it worthwhile to be further studied and clinically applied.

## 5. Conclusions

We note that the literature concerning GI and its proper treatment is limited. By presenting our case report and the other studies from the literature, we want to highlight to surgeons the importance of an early diagnosis as well as prompt treatment in order to preserve the viability of the penile glans and prevent potential complications. We also outline that in the presented case, various methods of treatment were tried before HBOT but without clinical success. In light of the presented cases, we conclude that in the case of acute ischemia of the penis glans, hyperbaric oxygen therapy (HBOT) is the treatment of choice, especially when other treatments do not work. Taking into consideration that these are very rare cases and the lack of specialized guidelines to treat acute penis glans ischemia, we outline the importance of further multi-center studies to establish accurate risk factors and management guidelines. Moreover, studies that could determine ethological factors leading to this rare complication that could affect penile functions in pediatric patients as well as adult patients should be a research direction in order to help practitioners apply a proper treatment in acute ischemia and achieve success in restoring normal function of the organ.

## Figures and Tables

**Figure 1 jpm-13-01370-f001:**
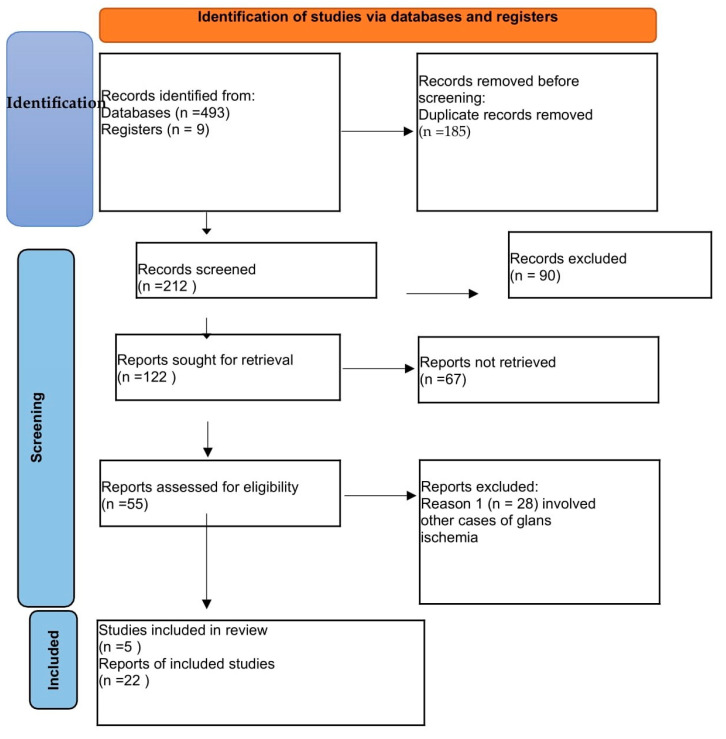
Prisma flow diagram.

**Figure 2 jpm-13-01370-f002:**
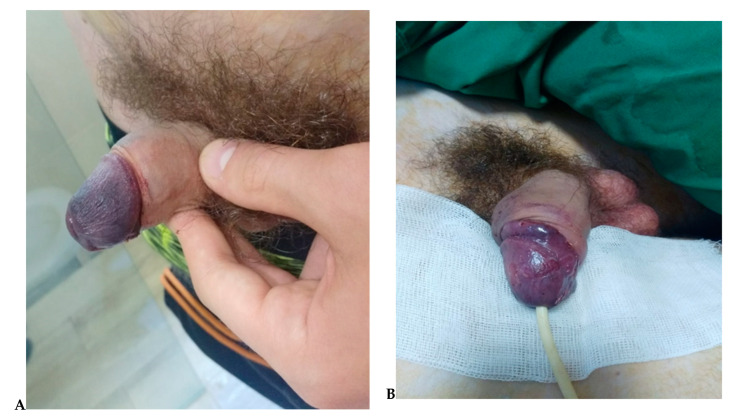
(**A**)—Glans aspect at five hours after surgery. We note the necrotic appearance of the glans. (**B**)—Glans aspect 48 h: it maintained the aspect; no improvements were noticed after treatment.

**Figure 3 jpm-13-01370-f003:**
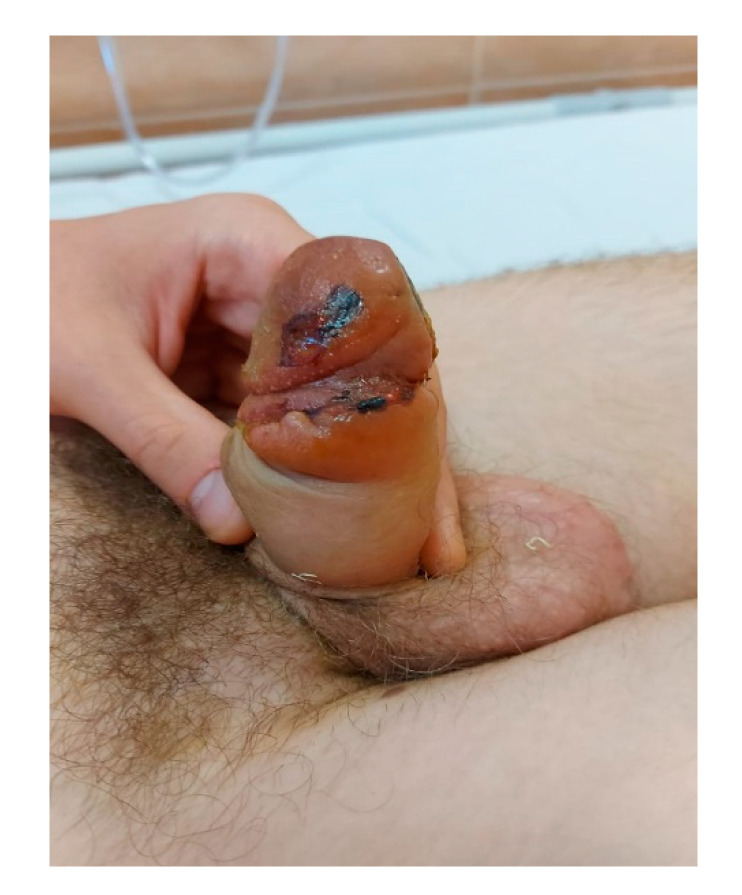
Penile glans aspect after first session of HBOT. We note the vascular flux restoration after the first session.

**Figure 4 jpm-13-01370-f004:**
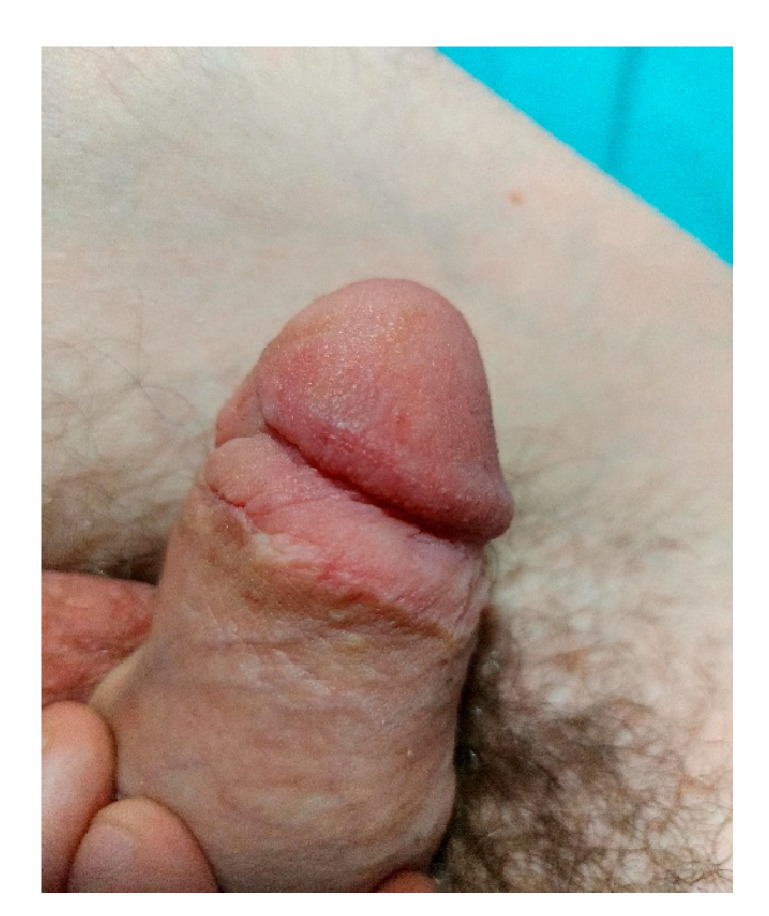
Penile glans at one month after surgery.

**Table 1 jpm-13-01370-t001:** Authors that included HBOT treatment for glans ischemia after circumcision.

Lead Author	Year	Patients Number	Age (Year)	Treatment Method
Polak N. [17]	2020	2 patients	Neonates	Treated via monotherapy with HBOT
Migliorini F. [6]	2018	1 patient	24 years old	Treated with association of HBOT and (pentoxifylline) PTX
Pepe P. [19]	2015	1 patient	unk	Treated with HBOT and corticoid therapy
Elemen L. [25]	2012	1 patient	unk	Treated with association of HBOT and PTX for vasodilation
Tzeng Y.S. [18]	2004	1 patient	33 years old	Treated with association of HBOT and PTX

## Data Availability

The data presented in this study are available on request from the corresponding author.

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
