# Peer review of "Severe Acute Ischemia of Glans Penis after Achieving Treatment with Only Hyperbaric Oxygen Therapy: A Rare Case Report and Systematic Literature Review"

_jpm, 2023, doi:10.3390/jpm13091370_

Round 1

Reviewer 1 Report

This potentially sounds as an interesting case report, as rarely presented in peer-reviewed literature to the date.

Nevertheless it represents a case of acute ischemia, a general indication already documented as a valid candidate to an adiuvant hyperbaric oxygen therapy.

With the due respect, even if I'm not qualified as an english native-speaker could be, on my opinion the present version of Authors' submission seems to need a major revision of the text, as there are frequent grammar, spelling, punctuation, and other writing/graphic issues.

It is proposed to adopt more attention in the graphic presentation of the systematic review (text orientation/spacing; repeated brackets to be removed) and accuracy in the bibliographic references (for example: Migliorini F. and not Filippo M; see at line 130). The Bibliography references should be reformatted unambiguously and accordingly to the requests of the journal, with the appropriate numerical references in the text (just in example, they're lacking in numerical references for Polak et al., at line 124, Migliorini et al., at line 130 ... and so on).

There are overlapping photos (at page 7 out of 12) in the *.pdf version of the article.

Please review extensively the text; the following are some annotations, even if only for lines 1-39:

    *** Line           *** Please correct

009           check font pitch at "540139"

017           reviewed., >> reviewed;

022           removing extra space between "space of ... HBOT."

025           removing extra space between "that HBOT ... is the treatment of choice"

026           when other treatment it does not work >> when other treatments do not work

033           patients[1,2]. >>  patients [1,2].

034           removing extra space between "sThe most ... common complications."

035-037       The special cases of complications in hemophiliacs deserve special mention,circumcision in these cases may increas the potential of complications [1,2,3]. >> The possible complications of such procedure in hemophilic patients are worthy of a particular mention as circumcision may increase the potential of complications in these cases [1,2,3]. 

Author Response

It is proposed to adopt more attention in the graphic presentation of the systematic review (text orientation/spacing; repeated brackets to be removed) and accuracy in the bibliographic references (for example: Migliorini F. and not Filippo M; see at line 130). The Bibliography references should be reformatted unambiguously and accordingly to the requests of the journal, with the appropriate numerical references in the text (just in example, they're lacking in numerical references for Polak et al., at line 124, Migliorini et al., at line 130 ... and so on).

There are overlapping photos (at page 7 out of 12) in the *.pdf version of the article.

We thank you very much for this observation, we revised the references and mistakes.

Comments on the Quality of English Language

Please review extensively the text; the following are some annotations, even if only for lines 1-39:

    *** Line           *** Please correct

009           check font pitch at "540139"

017           reviewed., >> reviewed;

022           removing extra space between "space of ... HBOT."

025           removing extra space between "that HBOT ... is the treatment of choice"

026           when other treatment it does not work >> when other treatments do not work

033           patients[1,2]. >>  patients [1,2].

034           removing extra space between "sThe most ... common complications."

035-037       The special cases of complications in hemophiliacs deserve special mention,circumcision in these cases may increas the potential of complications [1,2,3]. >> The possible complications of such procedure in hemophilic patients are worthy of a particular mention as circumcision may increase the potential of complications in these cases [1,2,3]. 

We appreciate the distinguished reviewers’ observation in this regard, we revised the mistakes and introduced a detetalied paragraph about possible complications in hemophiliacs.

Reviewer 2 Report

This is an overall very interesting case report about a severe complication after circumcision, and about successful HBOT in patient treatment.

However, there are some points that need revision:

At first, I would recommend to present the case report first, and the literature review second.

In introduction and discussion there are many repetitions of already given information. Please, revise.

L 104: should be comorbidities

L 108: median?? Should be deleted

Ll 118-121: sentence needs to be revised, is inconclusive.

L 130: introduce PTX as Pentoxifyllin first.

L 242: what is CDE?

L 205: I am confused about HBOT with 80kPa O2-pressure. Normal ambient partial pressure of O2 is already 20kPa (20% of 100kPa air), which means 80kPa of oxygen equals an Fi of only 80% O2…

In usual HBOT you treat with 240kPa O2, which equals 2.4 bar of pressure at 100% O2.

A spell-check for English language should be used.

Author Response

This is an overall very interesting case report about a severe complication after circumcision, and about successful HBOT in patient treatment.

However, there are some points that need revision:

At first, I would recommend to present the case report first, and the literature review second.

In introduction and discussion there are many repetitions of already given information. Please, revise.

 We thank the distinguished reviewer for the appreciation and for raising this important question, we revised the raised issues.  We thank the distinguished reviewer for this observation and suggestion, we revised the issues and we also explained what CDE is - Color Doppler Echography.

Reviewer 3 Report

Thank you for considering me as the reviewer of the manuscript titled “Severe acute ischemia of glans penis post achieving treatment only with hyperbaric oxygen therapy: a rare case report and literature systematic review”.

There are some comments that should be addressed:

The presentation of case is well written. However, the authors should state about any follow up visit of this patient. Normal function of the penis should be reported.

Please discuss the etiology of glans ischemia in discussion section.

What was the date of search?

What were the inclusion and exclusion criteria for systematic review?

Present the full search strategies for all databases.

Please revise the name of Figure 1. It is PRISMA flow diagram, and not a search strategy.

In the text, the authors stated, “A total of 55 articles were considered as eligible, 27 articles were analyzed in the review; 22 were original case 110 reports and 5 were reviews”. However, in PRISMA it is otherwise.

We don’t include review studies in systematic review.

Specify the methods used to assess risk of bias in the included studies

Minor editing of English language required

Author Response

The presentation of case is well written. However, the authors should state about any follow up visit of this patient. Normal function of the penis should be reported.

We appreciate the thoroughness of the distinguished reviewer’s evaluation. We have proofread the manuscript, and have corrected all formatting errors throughout it.

Please discuss the etiology of glans ischemia in discussion section.

We thank the distinguished reviewer for raising this issue in need of clarification, we included a paragraph of the etiology of glans  

What was the date of search?

We thank the distinguished reviewer for raising, we included the date of search.

What were the inclusion and exclusion criteria for systematic review?

We thank the distinguished reviewer for raising this important question. We included the inclusion and exclusion criteria.

Present the full search strategies for all databases.

We thank the distinguished reviewer for the appreciation and provided suggestions.  We delatied the search strategies for all databases.

Please revise the name of Figure 1. It is PRISMA flow diagram, and not a search strategy.In the text, the authors stated, “A total of 55 articles were considered as eligible, 27 articles were analyzed in the review; 22 were original case 110 reports and 5 were reviews”. However, in PRISMA it is otherwise. We don’t include review studies in systematic review.

We thank the distinguished reviewer for this observation. We revised this issue.

Specify the methods used to assess risk of bias in the included studies

We highly appreciate the distinguished reviewer’s observation in this regard. We detailed how we assessed the risk of bias.

Round 2

Reviewer 1 Report

Interesting work that I believe still requires a small effort of accuracy in some points (see file *: zip):

a) At page # 4: I suggest again to rebuild the Prisma flow diagram so to standardize the characters/spaces/orientation of the text (only an approximate example in the image present into the *.zip file attached).

b) At page # 5: Unfortunately another "Filippo M." to be changed in "Migliorini F." (as per note reported in the *.pdf You'll find in the compressed file).

c) Useful a last English language revision, just to avoid leaving further information that the Authors wanted to share on the case ... and, in my opinion, They will have been perfectly successful in order to present their work in a fluid and very clear manner.

Best regards

Author Response

We thank the distinguished reviewer for the appreciation and provided suggestions, we revised the manuscript. 

Reviewer 3 Report

All of my comments have been responded by the authors.

English language fine. No issues detected

Author Response

We thank the distinguished reviewer for the appreciation and support.